# Exploring the impact of nature connectedness on well-being and mental distress among urban youth: Evidence from 25 most populated cities in India

**Raina Chhajer**[1], **Sudeep R. Bapat**[2]*

**1** Humanities and Social Sciences Area, Indian Institute of Management Indore, Indore, India, **2** Shailesh J. Mehta School of Management, Indian Institute of Technology Bombay, Mumbai, India

* sudeepb@iitb.ac.in

**Data availability statement:** The data is available at https://doi.org/10.6084/m9.figshare.27612366.

## Abstract

In contemporary urban settings, enhancing the well-being of youth is a challenge. The proliferation of screen time, particularly post-pandemic, has contributed to an alarming disconnection with nature. This study investigates the impact of nature connectedness on the well-being of urban youth residing in the 25 most populated cities in India. 2283 urban youth responded to an online survey questionnaire that aimed to assess participants' exposure to and interaction with nature. Respondents self-reported their proximity to green and blue spaces near their residences, the frequency of recreational visits to these areas, time spent in indoor environments, nature connectedness, and well-being. Multivariate regression analysis results show that urban youth with access to both green and blue spaces within 1-kilometre or between 1-3 kilometres radius, along with frequent recreational visits to these spaces, reported a significant positive impact on well-being. Additionally, spending less than 8 hours in indoor environments and a high nature connectedness score had a significant positive impact on well-being. Moreover, a logistic regression analysis shows that urban youth with well-being score less than 13, considered mentally distressed, who have access to both green and blue spaces within 1 kilometres or access to green spaces between 1-3 kilometres, along with frequent recreational visits to these spaces, and a high nature connectedness score, increases their log-odds of well-being significantly. This study conducted in the 25 most populated cities, represents a comprehensive exploration of the impact of nature connectedness on well-being in India. The study does, however, have limitations, including uneven sample distribution among cities, the absence of temporal analysis, and a restricted age range of 18-25 years. These limitations will be addressed in future research. The implications of the study extend to policy recommendations, advocating for the development of more parks in close proximity to residential areas. Such initiatives aim to encourage urban youth to actively engage with and experience nature, fostering improved well-being and reduced distress levels.

**Funding:** The author(s) received no specific funding for this work.

**Competing interests:** The authors have declared that no competing interests exist.

## Introduction

Mental distress among urban youth has been consistently increasing over the years. In America, India and European countries, urban youth face common mental health disorders [1], high levels of depressive symptoms [2], and higher prevalence of psychosis, mood and anxiety disorders [3] when compared to their rural counterparts. In India, it was shown that urban students were more adversely affected by COVID-19 than those living in rural areas [4], and a vast range of youth suffer from various mental health problems [5]. The problems arise due to a variety of factors, including lifestyle factors such as age, area of residence, income and occupation [5], social factors such as relationship issues, social mores and academic pressures [6], and external factors such as exposure to violence [7,8]. Urbanization has also been speculated to be a reason for the same, as the rural youth report higher satisfaction with material and spiritual well-being than their urban counterparts [9]. Mental health issues affect the daily lives of urban youth in many ways, as they lower levels of academic achievement and productivity [2], increase substance abuse [3], and decrease overall satisfaction with life [10]. Several environmental techniques, such as activities in urban or natural environments, are perceived to positively affect emotion regulation and enhance mental health [11]. Hence, having a higher sense of connection with nature has become extremely important in today's world.

According to Nisbet and Zelenski, 'Nature [connectedness or] relatedness refers to the subjective sense of connection people have with the natural environment' [12]. Natural connection includes an individual's sense of belonging and affinity for the natural world, including their awareness, emotional connect, and actions that builds the human-nature relationship. A higher sense of connection with nature is essential for overall well-being, as nature has repeatedly been shown to enhance the quality of life in individuals. Proximity to natural elements such as greenery [13] or coast view [3,14] has been shown to enhance mental well-being. Even exposure to nature in virtual reality showed improvements in stress reduction, relaxation, mood, attention, nature connection, and nature spirituality in urban adolescents and college students [15,16].

More than proximity and exposure, direct engagement with nature has been shown to tremendously impact mental and physical health. When allowed to interact with nature outdoors, preschool children revealed that their overall physical health improved, along with their moods and reduced stress and anger outbursts [17]. Studies across Australia and Europe showed that more extended visits to nature were associated with lower depression, higher blood pressure [18], more positive engagements with neighbours, and enhanced well-being [19]. Performing activities in nature has been repeatedly shown to improve connection with nature, well-being [20], attention capacity, and the ability to reflect on life problems [21]. Stress reduction interventions, such as Mindfulness-Based Stress Reduction, had greater positive effects on mental health and well-being when conducted outdoors in a natural environment than indoors [22]. Different outdoor activities in nature also help families grow stronger bonds and become closer [23]. When probed, undergraduate students revealed that they tended to establish identities with themselves, their attitudes, and nature; forming a balanced triadic structure of implicit environmental identity [24]. During the COVID-19 pandemic, people visited nature more often and took to gardening [25], perceiving nature as an additional resource to improve their mental health and well-being [26].

This leads to our next question: Why is connecting with nature beneficial? Several theories have explored this concept. One such hypothesis, the Biophilia hypothesis, suggests that humans have an innate inclination to connect with nature, which stems from evolutionary survival mechanisms. This intrinsic bond explains why spending time in the natural

environment can evoke feelings of peace, fulfilment, and belonging. [27]. In essence, since humans have developed primarily in natural environments, they are psychologically better off in the same compared to urban settings [29]. Stress reduction theory further supports this by emphasizing that simply observing greenery, water bodies, or other natural elements can trigger physiological relaxation responses, reducing cortisol levels, and promoting emotional stability. [28]. When individuals immerse themselves in nature, be it walking in a forest, listening to birds chirping, or feeling the breeze, there is a noticeable decline in stress and anxiety. This calming effect of nature has been widely observed across cultures and age groups, reinforcing its universal significance for human health.

Additionally, the attention restoration theory posits that exposure to natural environments replenishes mental energy and improves concentration. In fast-paced urban settings, individuals often experience cognitive fatigue due to constant demands on their directed attention, such as digital distractions, work responsibilities, and social commitments. Nature, in contrast, provides a form of effortless attention, where the mind can wander and recover without strain [30].

Baxter and Pelletier say that 'there is evidence of nature-relatedness having a positive impact on human health and functioning in three major areas: stress-related physiology and recovery, psychological well-being, and cognitive recovery from response inhibition and attentional fatigue' [31]. Nature-based activities help reduce anxiety levels and affect perceived restoration due to the development of a feeling of nature connectedness [32], improving overall well-being and self-control [33]. Short-term exposure to green spaces can reduce stress and depressive symptoms and increase self-esteem, mood, mental and physical health, and the development of nature connectedness in undergraduate students [34,35]. Moreover, long-term exposure reduces mortality and improves overall well-being [35,36]. Engagement with nature by visiting more than once a week can lead to better health, prevention, and treatment of many public health challenges, such as obesity, heart disease, depression, and anxiety [37–39]. It has been found that people with a stronger connection to nature were less likely to be depressed, stressed, and anxious [40], and tend to be happier, have higher attention spans, encounter less cognitive fatigue [41], have an overall improved health, attitudes and behaviours [42], have lower state and trait cognitive anxiety [43]. Regarding youth, college students benefit from regular walks in nature in terms of mental health, learning engagement, attention recovery, reflection experiences, and connectedness with nature [44]. Nature interventions have increased happiness and connectedness among university students [45]. In children, nature helped buffer the effects of stressful life events such as bullying, family relocation, punishments, and peer pressure [46].

These results are consistent when different geographic locations are also considered. A meta-analysis covering several studies from Canada, Europe, the United States, Australia, India, Colombia, and Hong Kong revealed that individuals connected to nature tend to be better off psychologically, as nature connectedness was positively correlated with eudemonic well-being [39]. This was specifically true for personal growth, which had a stronger relationship with nature connectedness than most other factors. Nature connectedness positively correlates with psychological and social well-being [47] and is considered a distinct and positive predictor of happiness [48] in students in Canada and the US. Another study, conducted across 14 European countries, USA, China, Canada, and Australia, also showed similar results [49]. In Australia and the UK, nature connectedness and engagement with nature had a stronger impact on well-being than time spent outdoors [50,51], and people participating in outdoor physical activities demonstrated lower somatic anxiety levels and felt more connected to nature than those participating in indoor physical activities [52]. In China, simple tasks, such as taking care of houseplants, induced greater levels of well-being and mindfulness

traits in adults [53], while urban Chinese youth displayed enhancement in cognitive reappraisal and reduction in expressive suppression when in direct or indirect contact with nature [54]. In Southern England, time spent in private green spaces positively affected mental and social health while increasing nature orientation and physical activity and reducing symptoms of depression [55]. In India, studies indicate that a higher frequency of nature visits leads to higher mindfulness in adults [56], and nature connectedness positively affects subjective happiness and builds resilience [57]. Nature-relatedness also improved mental well-being in Indian adults and helped them cope with the effects of the COVID-19 pandemic [58].

Research on nature connectedness and well-being among urban youth in India is still in it's infancy and limited to small samples. The current study aims to fill this gap by analysing data on residential exposure and self-reported frequency of recreational visits to green and blue spaces from 25 cities in India, with an overall sample of 2283 respondents. Data regarding the respondents' well-being was collected and measured using the World Health Organisation's 5-item index of well-being (WHO-5) [59]. Self-reported time spent indoors was also recorded. Finally, the Inclusion of Nature in Self (INS) scale was used to measure their psychological connectedness to the natural environment [60].

This paper addresses how residential exposure to green and blue spaces, recreational visits, time spent indoors, nature connectedness, and frequency of recreational visits to local and other city-based natural environments influence well-being and mental distress outcomes. Further, how do these factors vary across different cities, and what is the relationship between city-specific environmental characteristics and mental health indicators? These questions led to four hypotheses (H). H1: Greater residential exposure, measured by the proximity to green and/or blue spaces, will be associated with (a) higher positive well-being and (b) lower probability of mental distress. H2: More frequent recreational visits to green and/or blue spaces will lead to the above relations for residential exposure. H3: Time spent indoors will significantly predict the two outcomes. Finally, H4: Nature connectedness, will be a significant independent predictor of well-being and mental distress outcomes. Although not the primary focus of the study, a city-specific analysis was also conducted as an additional exploration to examine the relationships between the variables across different cities.

## Materials and methods

### Sample and survey

Data for this study was collected from urban youth (aged 18-25) residing in the 25 most populous cities in India, focusing on their exposure to nature through access to green and blue spaces near their residences. The online survey was administered in January 2024. Responses were gathered from the following cities, ranked by population size: Mumbai, Delhi, Bangalore, Hyderabad, Ahmedabad, Chennai, Kolkata, Surat, Pune, Jaipur, Lucknow, Kanpur, Nagpur, Indore, Bhopal, Vishakhapatnam, Patna, Vadodara, Ludhiana, Agra, Nashik, Rajkot, Srinagar, Aurangabad, and Kochi. The distribution of cities across the country helps ensure a diverse and representative sample.

Using convenience sampling method data was collected through academic institutions, social media platforms, and local community networks to maximize reach and accessibility. The survey was conducted online to facilitate wide participation, and ethical considerations were strictly adhered to, ensuring informed consent, confidentiality, and anonymity for all participants. A total of 2,410 responses were initially received, with incomplete data removed to finalize the sample at 2,283 participants.

The questionnaire was designed using psychometrically validated tools, including the WHO-5 well-being index and the Inclusion of Nature in Self Scale to measure nature

connectedness. Minimal language adaptation was made to ensure cultural relevance, and a pilot test was conducted to refine questions and resolve any ambiguities. To minimize potential biases, measures such as ensuring anonymity, neutral question wording, and randomized item order were implemented. Standardized instructions were provided to ensure uniform understanding across participants.

To mitigate survey fatigue, we took several measures—the survey was designed to be concise, focusing on key variables while minimizing the number of questions. We structured the survey in a logical and engaging manner to maintain participant interest throughout. Additionally, during the pilot survey, we checked if the length and complexity were manageable for the participants. These steps were aimed at reducing the likelihood of survey fatigue. The study procedure was approved by the Institutional Review Board (reference number IRB/05/2023-24/HSS) and conducted in compliance with the Declaration of Helsinki (2013 revision).

## Residential exposure

The participants were asked if they had access to nature, i.e., either a green or blue space around their residence. A green space is generally defined as an area containing grass, trees or other vegetation used for recreational or aesthetic purposes, whereas a blue space is defined as a body containing natural or man made surface water. The participants could answer if they had access to 'only green space', 'only blue space', 'both green and blue spaces' or neither of them. Further, they were specifically asked whether they had access to a green or blue space near their residence and in what capacity. The participants had to form a rough idea about the distance between a green or blue space and their residence. They could input 'not at all', 'within 1 Km', '1–3 Km', '3–5 Km', or 'more than 5 Km' as their responses.

## Recreational visits

The participants were asked multiple questions based on their interaction with nature, especially for a recreational purpose. A basic question asked whether they visited a green or blue space for recreational purposes, for which they could answer either a yes or no. Further, the participants who answered a yes to the previous question were also asked the purpose of their visit, which could be either 'for relaxation', 'spending time with family and friends', undertaking physical activities such as exercising' or 'anything else', and their frequency of such visits, which could be 'once every month', 'once every three months' or 'once every six months'. Fig 1 contains a barplot showing the distribution of recreational purpose across recreational frequency. The category 'once every month' is seen to be predominant across different recreational purposes. The second set of questions which they were asked was whether they visited other cities for nature based recreation, for which they could answer either a yes or no. The participants who answered a yes to the previous question were then asked their frequency of such visits, where the response categories were same as before.

## Well-being

We adopted the five item WHO-5 well-being Index [59]. Respondents had to provide answers to 5 questions using a six-point Likert scale with the following categories: 0 (at no time), 1 (some of the time), 2 (less than half the time), 3 (more than half the time), 4 (most of the time), to 5 (all of the time). The individual scores were summed to get a total score out of 25. A higher score indicated higher level of well-being, while total scores below 13 indicated mental distress.

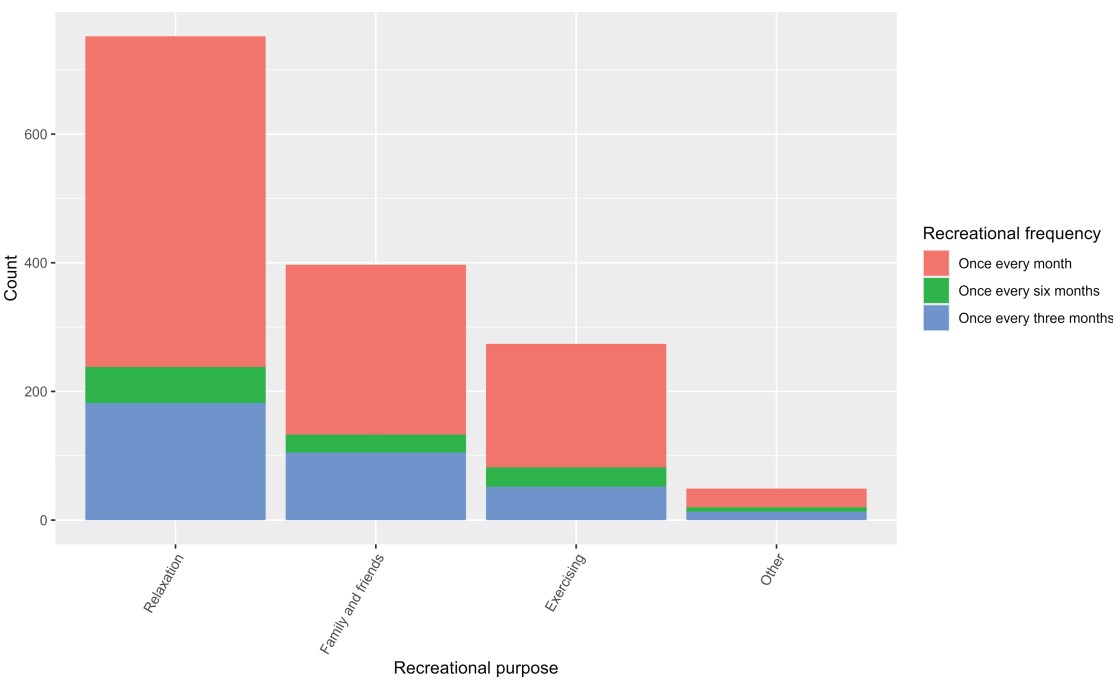

**Fig 1. Recreational purpose across different recreational frequencies.**

## Nature connectedness

Schultz's Nature in Self (NIS) [60] is a simple one-item measure with graphical representations that depends on self-reported answers. When repeated one or four weeks later, test-retest correlations for the NIS have shown high reliability [60]. Each of the seven Venn diagrams in the NIS shows two circles labelled 'nature' and 'self', with different amounts of overlap. Participants were instructed to circle the illustration most accurately depicting their interaction with the environment [61]. Non-overlapping circles indicated lowest nature connectedness, while most overlapping circles indicated highest nature connectedness.

## Covariates

The covariates in the study included basic demographic features such as gender (female = reference, male), age (18–25 years), education (high school = reference, graduate, post-graduate, or professional), and profession (student = reference, working professional), along with specific related variables such as time spent indoors, which denotes the number of hours spent by the person indoors (<4 Hrs = reference, 4–8 Hrs, 8–12 Hrs, >12 Hrs), residential exposure (neither green nor blue space = reference, only green space, only blue space, both green and blue spaces), access to green space near the residence (not at all = reference, <1 Km, 1–3 Km, 3–5 Km, >5 Km), access to blue space near residence (not at all = reference, <1 Kms, 1–3 Km, 3–5 Km, >5 Km), visiting green or blue spaces for recreation (Yes, No = reference), and nature connectedness (NC), explained through the 'Nature in Self' (NIS) scale (an ordinal variable from 0–6). These covariates were selected based on their established impact on well-being, as highlighted in prior literature. Fig 2 shows barplots depicting the distribution of categories among each of the covariates.

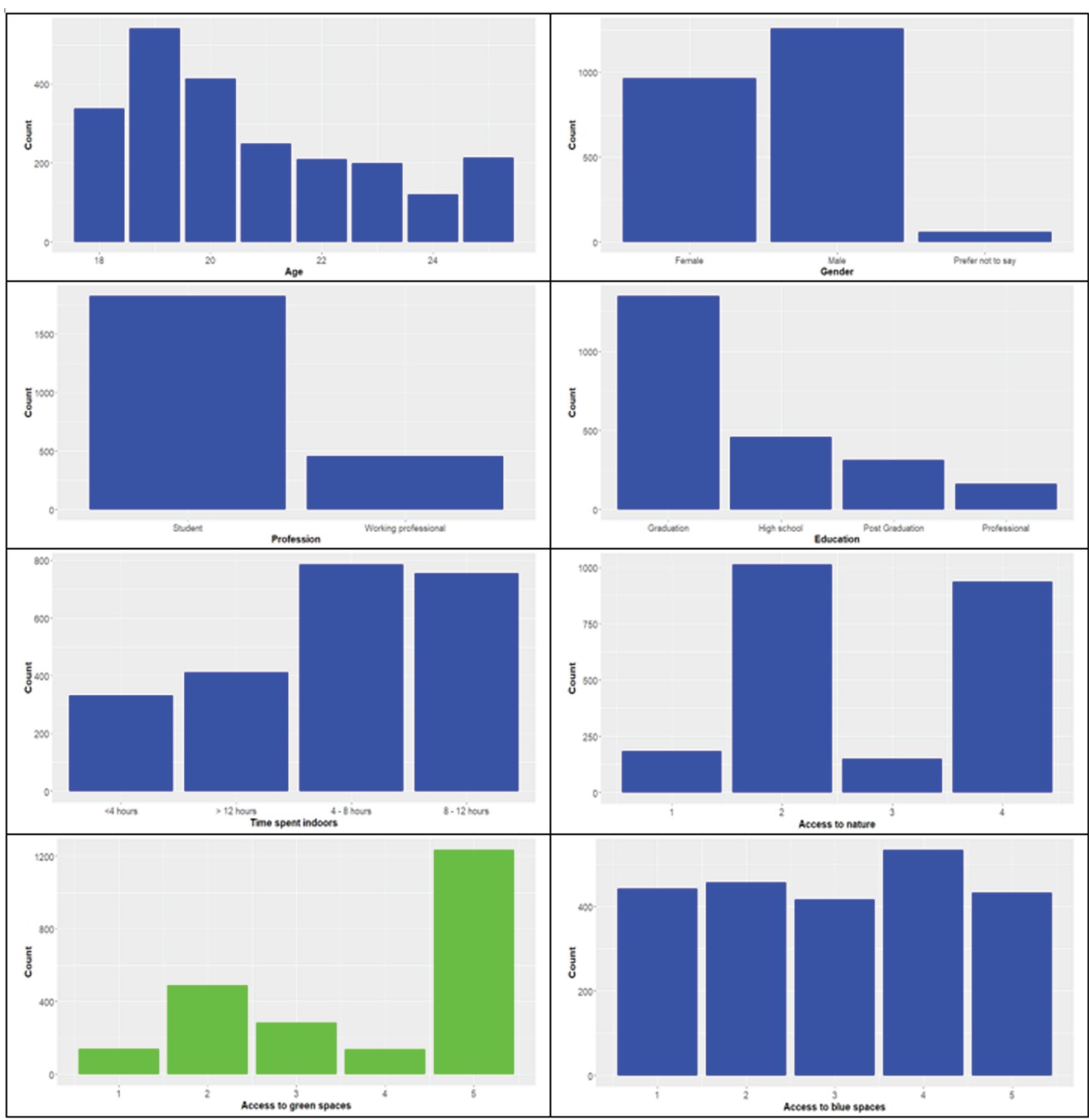

**Fig 2. Barplots showing the distribution of categories among the covariates.**

## Analyses

For this study, we adopted the standardized versions of the WHO-5 well-being index and the Inclusion of Nature in Self (NIS) scale without modification. These scales are widely used and have been validated across various international settings, ensuring their robustness in measuring well-being and nature connectedness. To confirm their applicability in the Indian

context, we assessed the internal consistency and reliability of both scales using Cronbach's alpha. The results indicated acceptable levels of reliability (Cronbach's alpha >0.7) for both scales, demonstrating their suitability for use in our study. The proposed hypotheses were tested using a series of multiple linear regression (MLR) models, for hypotheses where the response variable was WHO-5 score, and a suitable logistic regression model for capturing mental distress, where the response was a binary outcome. A person is prone to mental distress if the WHO-5 score is less than 13. These analyses assume that the above outlined covariates are held fixed. Further, a couple of sub-MLR models were run, keeping the response variables same as before. The first sub-model was for the group which visited the green or blue spaces for recreation purposes. The visits were mainly either for relaxation, spending time with family and friends, undertaking physical activities such as exercising, or something else. The covariates were recreational frequency (once every month, once every three months, once every six months = reference), residential exposure, access to green space, access to blue space and NIS. A city-wise comparison of the WHO scores for the 25 chosen cities was also done. The second sub-model was for the group which visited other cities for nature based recreation. The covariates were recreational frequency and NIS.

## Results

The descriptive summary for some of the important (significant) predictors is presented in Table 1. Participants reported moderate levels of nature connectedness (mean NIS score: 3.17, SD = 1.45) and well-being (mean WHO-5 score: approximately 17, SD = 5). Table 2 presents the relevant regression model summaries for the two sets of models (MLR and logistic). Note that for the mental distress variable, since we are taking the average score, a person whose score is <13 is termed as potential candidate suffering from mental distress. Table 3 contains results for the sub-model for the "recreational frequency" variable, whereas Table 4 contains results for the sub-model related to "recreational frequency to other cities".

### Residential exposure

The first hypothesis related to measuring the impact of residential exposure (proximity to green and/or blue spaces) on well-being, can be supported through the following observations from Table 2. A person having both green and blue spaces around his residence is seen to significantly impact his well-being ($p = 0.0966 < 0.1$). The corresponding $\beta$ coefficient is 0.83 which signifies that if a person has access to both green and blue spaces around his residence, as opposed to having none, the WHO-5 score increases by 0.83, keeping all the other covariates constant. Further, a person having a green space either within 1 Km ($p = 0.00013$, $\beta = 2.18$), between 1–3 Km ($p = 0.019 < 0.05$, $\beta = 1.33$), or between 3–5 Km ($p = 0.077 < 0.1$, $\beta = 1.03$) is also seen to significantly positively impact his well-being. One can specifically conclude about the significance using the $\beta$ coefficient as done before.

On the other hand, one can also see that having access to a green space within 1 Km (OR = 0.33, $\beta = –1.11$, $p = 0.001$), between 1–3 Km (OR = 0.47, $\beta = –0.75$, $p = 0.012 < 0.05$), or between 3–5 Km (OR = 0.58, $\beta = –0.55$, $p = 0.073 < 0.1$) of the residence is seen to be negatively associated with the likelihood of mental distress (since all the coefficients are negative). Specifically, if a person has access to a green space within 1 Km, between 1–3 Km or between 3–5 Km of his residence, as opposed to none, the chance of observing mental distress reduces. Also, the log-odds of being mentally distressed decreases by 1.11,0.75,0.55 respectively for the three groups.

**Table 1. Summary statistics for the WHO score and mental distress, based on the predictors.**

| | | | WHO-5 | | | WHO-5 < 13 | |
|---|---|---|---|---|---|---|---|
| Predictor | Count | % | Mean | SD | Count | % | Mean |
| **Gender** | | | | | | | |
| Female | 964 | 42.24 | 17.72 | 5.36 | 159 | 6.97 | |
| Male | 1259 | 55.17 | 17.26 | 5.22 | 219 | 9.6 | |
| **Education** | | | | | | | |
| Graduation | 1351 | 59.20 | 17.58 | 5.29 | 227 | 9.95 | |
| High school | 458 | 20.07 | 17.26 | 5.35 | 82 | 3.59 | |
| Post Graduation | 312 | 13.67 | 16.97 | 5.17 | 60 | 2.63 | |
| Professional | 161 | 7.06 | 16.87 | 5.62 | 34 | 1.49 | |
| **Profession** | | | | | | | |
| Student | 1827 | 80.06 | 17.32 | 5.28 | 322 | 14.11 | |
| Working professional | 455 | 19.94 | 17.62 | 5.46 | 81 | 3.55 | |
| **Residential exposure** | | | | | | | |
| Neither green nor blue space | 184 | 8.06 | 14.03 | 6.14 | 68 | 2.98 | |
| Both green and blue spaces | 1014 | 44.43 | 18.46 | 5.14 | 130 | 5.7 | |
| Only blue space | 148 | 6.49 | 14.95 | 5.08 | 46 | 2.02 | |
| Only green space | 936 | 41.02 | 17.24 | 4.94 | 159 | 6.97 | |
| **Access to green space** | | | | | | | |
| Within 1 Km | 1235 | 54.12 | 18.47 | 5.05 | 151 | 6.62 | |
| 1–3 Kms | 490 | 21.47 | 16.97 | 4.89 | 85 | 3.72 | |
| 3–5 Kms | 284 | 12.45 | 16.14 | 4.94 | 65 | 2.85 | |
| >5 Kms | 138 | 6.05 | 15.41 | 5.39 | 44 | 1.93 | |
| Not at all | 135 | 5.92 | 13.47 | 6.55 | 58 | 2.54 | |
| **Access to blue space** | | | | | | | |
| Within 1 Km | 432 | 18.93 | 18.49 | 5.88 | 75 | 3.29 | |
| 1–3 Km | 457 | 20.03 | 17.46 | 4.99 | 68 | 2.98 | |
| 3–5 Km | 417 | 18.27 | 16.76 | 5.17 | 83 | 3.64 | |
| >5 Km | 442 | 19.37 | 17.83 | 4.53 | 54 | 2.37 | |
| Not at all | 534 | 23.40 | 16.51 | 5.62 | 123 | 5.39 | |
| **Recreational visits** | | | | | | | |
| No | 650 | 28.48 | 14.79 | 5.10 | 195 | 8.55 | |
| Yes | 1632 | 71.52 | 18.41 | 5.04 | 208 | 9.11 | |
| **Time spent indoors** | | | | | | | |
| 4–8 hours | 786 | 34.44 | 17.32 | 5.05 | 135 | 5.92 | |
| 8–12 hours | 754 | 33.04 | 17.27 | 4.81 | 119 | 5.21 | |
| <4 hours | 331 | 14.50 | 18.47 | 5.99 | 54 | 2.37 | |
| >12 hours | 411 | 18.01 | 16.82 | 5.97 | 95 | 4.16 | |
| **Nature connectedness** | | | | | | | |
| NC | - | - | 3.17 | 1.45 | - | - | 2.20 |

## Recreational visits

The second hypothesis related to measuring the impact of recreational visits (visiting green or blue spaces for recreational purposes) on well-being, can be supported through the following observations from Table 2. A person visiting green or blue spaces for recreational purposes has a significant positive impact on his well-being ($p<0.0001$, $\beta = 1.69$, Table 2). Thus, the average WHO score increases by 1.69, is he visits such spaces as opposed to if he does not. Also, one can note that a person visiting green or blue spaces for recreational purposes is negatively associated with the chances of being mentally distressed (OR = 0.65, $\beta = -0.43$, $p = 0.0001$).

**Table 2. MLR and logistic regression summaries for the predictors.** $^{\#}p < 0.1,^{*} p < 0.05,^{**} p < 0.01.^{***}p < 0.001$

| Predictor | WHO-5 | | WHO-5 < 13 | |
|---|---|---|---|---|
| | Estimates | 95% CIs | Odds ratios | 95% CIs |
| Intercept | 9.59 | (7.86,11.32) | 5.24 | (1.76,15.90) |
| Age | −0.12 | (−0.24,0.01) | 1.01 | (0.94,1.09) |
| **Gender** | | | | |
| Male | −0.28 | (−0.68,−0.11) | 1.03 | (0.80,1.33) |
| **Education** | | | | |
| Graduation | 0.36 | (−0.14,0.86) | 0.92 | (0.68,1.27) |
| Post Graduation | −0.15 | (−0.93,0.64) | 1.07 | (0.6,1.72) |
| Professional | −0.10 | (−1.12,0.92) | 1.03 | (0.55,1.91) |
| **Profession** | | | | |
| Working professional | 0.47 | (−0.33,1.27) | 1.07 | (0.66,1.71) |
| **Residential exposure** | | | | |
| Both green and blue spaces | 0.83$^{\#}$ | (−0.15,1.82) | 0.81 | (0.47,1.41) |
| Only blue space | −0.35 | (−1.47,0.77) | 1.26 | (0.69,2.30) |
| Only green space | 0.02 | (−0.93,0.96) | 1.22 | (0.74,2.05) |
| **Access to green space** | | | | |
| Within 1 Km | 2.18$^{***}$ | (1.06,3.30) | 0.33$^{***}$ | (0.18,0.60) |
| 1–3 Kms | 1.33$^{*}$ | (0.21,2.44) | 0.47$^{*}$ | (0.26,0.85) |
| 3–5 Kms | 1.03$^{\#}$ | (−0.11,2.17) | 0.58$^{\#}$ | (0.32,1.05) |
| >5 Kms | 0.28 | (−0.93,1.49) | 1.04 | (0.56,1.93) |
| **Access to blue space** | | | | |
| Within 1 Km | −0.93$^{*}$ | (−1.68,−0.18) | 2.28$^{***}$ | (1.42,3.67) |
| 1–3 Kms | −0.72$^{*}$ | (−1.40,−0.04) | 1.13 | (0.74,1.73) |
| 3–5 Kms | −0.56 | (−1.23,0.11) | 1.22 | (0.81,1.83) |
| >5 Kms | 0.20 | (−0.41,0.82) | 0.70$^{\#}$ | (0.46,1.04) |
| **Recreational visits** | | | | |
| Yes | 1.69$^{***}$ | (1.23,2.16) | 0.65$^{**}$ | (0.50,0.84) |
| **Time spent indoors** | | | | |
| 4–8 hours | −0.81$^{**}$ | (−1.41,−0.21) | 1.09 | (0.70,1.62) |
| 8–12 hours | −0.76$^{*}$ | (−1.37,−0.16) | 0.75 | (0.64,1.42) |
| >12 hours | −0.40 | (−1.08,0.29) | 1.07 | (0.70,1.65) |
| **Nature connectedness** | | | | |
| NC | 1.37$^{***}$ | (1.24,1.51) | 0.59$^{***}$ | (0.54,0.64) |

## Time indoors

The third hypothesis related to the impact of time spent indoors on well-being, can be supported through the following observations from Table 2. A person who spends between 4–8 hours ($p = 0.0008, \beta = −0.81$, Table 2) or between 8–12 hours ($p = 0.013, \beta = −0.76$) indoors is seen to have a negative impact on his well-being. However, the time spent indoors is seen to have no significant impact on the likelihood of being mentally distressed.

## Nature connectedness

The fourth hypothesis related to the impact of nature connectedness on well-being can be seen through the following observations from Table 2. Nature connectedness was seen to be significantly positively related to well-being ($p<0.0001, \beta = 1.37$, Table 2). The $\beta$ coefficient signifies that for every unit increase in the nature connectedness score of a person, his well-being score improves by 1.37 on an average. Further, nature connectedness was seen to be significantly negatively associated with mental distress ($OR = 0.59, \beta = −0.53, p<0.0001$). Fig 3 shows the predicted WHO-5 scores from the MLR model and the predicted probability of mental distress from the logistic regression model, for different NIS scores. The dashed

**Table 3. MLR regression summary for the recreational frequency.** #$p < 0.1$,* $p < 0.05$,** $p < 0.01$.*** $p < 0.001$

| Predictor | WHO-5 | | WHO-5 < 13 | |
|---|---|---|---|---|
| | Estimates | 95% CIs | Odds ratios | 95% CIs |
| Intercept | 14.49 | (12.69,16.29) | 0.53 | (0.16,1.56) |
| **Recreational frequency** | | | | |
| Once every month | 0.47** | (0.30,1.35) | 0.80* | (0.46,1.47) |
| Once every three months | 0.26# | (0.11,1.27) | 0.96 | (0.53,1.80) |
| **Residential exposure** | | | | |
| Both green and blue spaces | −0.27 | (−1.71,1.17) | 1.59 | (0.64,4.45) |
| Only blue space | −0.83 | (−2.51,0.85) | 1.40 | (0.47,4.34) |
| Only green space | −1.01 | (−2.43,0.42) | 1.76 | (0.72,4.81) |
| **Access to green space** | | | | |
| Within 1 Km | 0.83 | (−0.88,2.54) | 0.68 | (0.23,2.24) |
| 1–3 Km | 0.12 | (−1.60,1.84) | 1.00 | (0.34,3.32) |
| 3–5 Km | −0.21 | (−1.96,1.53) | 0.89 | (0.29,2.99) |
| >5 Km | −1.17 | (−3.04,0.70) | 2.63 | (0.86,8.99) |
| **Access to blue space** | | | | |
| Within 1 Km | −1.02** | (−1.88,−0.17) | 2.03** | (1.13,3.66) |
| 1–3 Km | −0.54 | (−1.34,0.25) | 0.91 | (0.51,1.60) |
| 3–5 Km | −0.53 | (−1.34,0.27) | 1.08 | (0.62,1.85) |
| >5 Km | 0.20 | (−0.52,0.92) | 0.67 | (0.39,1.12) |
| **Nature connectedness** | | | | |
| NC | 1.40*** | (1.24,1.57) | 0.57*** | (0.49,0.64) |

**Table 4. MLR regression summary for the nature based recreational frequency to other cities.** #$p < 0.1$, *$p<0.05$, **$p<0.01$.*** $p<0.001$

| Predictor | WHO-5 | | WHO-5 < 13 | |
|---|---|---|---|---|
| | Estimates | 95% CIs | Odds ratios | 95% CIs |
| Intercept | 13.97 | (13.09,14.84) | 1.23 | (0.72,2.08) |
| **Recreational frequency** | | | | |
| Once every month | 0.77** | (0.51,1.57) | 0.47* | (0.29,1.18) |
| Once every three months | 0.28 | (0.15,1.28) | 0.61 | (0.52,1.26) |
| **Nature connectedness** | | | | |
| NC | 1.39*** | (1.21,1.56) | 0.55*** | (0.48,0.63) |

light blue curves show the 95% confidence intervals around the fit. As one can note, for a higher NIS score, the predicted WHO-5 score happens to be higher, whereas the predicted probability of mental distress reduces by a great extent.

## Recreation frequency

Since the second hypothesis concerning recreation was seen to be significant, we can now talk about the hypothesis of measuring the impact of frequency of recreational visits by people to green or blue spaces on their well-being. This covariate is divided into three categories (once every month, once every three months, and once every six months = reference). This hypothesis can be supported through the following observations from Table 3. A person who visits such spaces for recreational purposes frequently, say, once every month ($p = 0.0041$, $\beta = 0.47$) or once every three months ($p = 0.0711$, $\beta = 0.26$) as opposed to once every six months, is seen to have a higher WHO-5 score, due to the positive $\beta$ coefficients. This is in line with our belief that more frequent visits to such spaces for recreational purposes improves their well-being. Further, a person who visits such spaces for recreational purposes once every month (OR

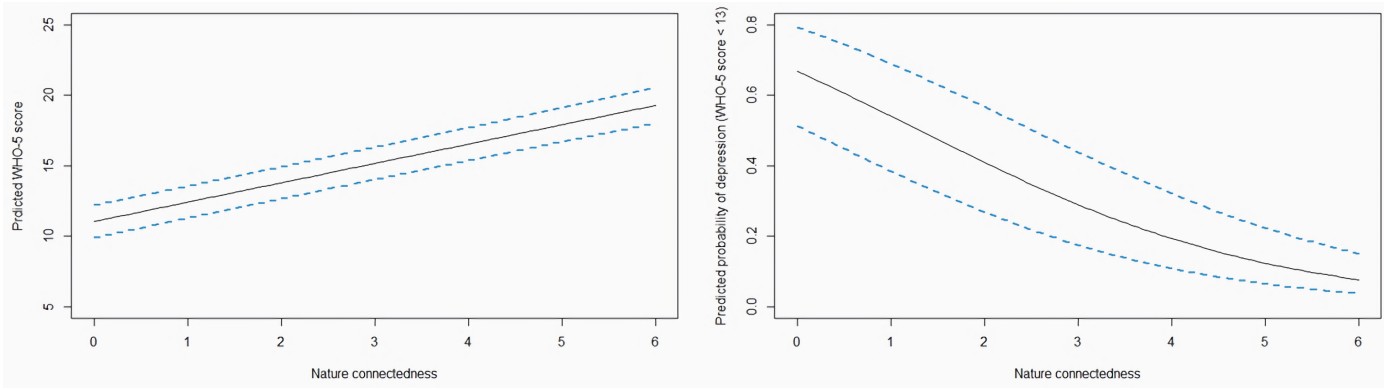

**Fig 3. Relationships between well-being and nature connectedness.** The plots are based on the predicted values from the MLR and logistic regression models assuming all the covariates (age, gender, education, profession, residential exposure, access to green and blue spaces, recreational visits, time spent indoors) are held fixed at their base levels.

= 0.80, $p$ = 0.0275) is seen to have lesser chances of acquiring mental distress as opposed to a person who visits once every six months.

## Recreation frequency in other cities

The hypothesis related to the measuring the impact of frequency of nature based recreational visits to other cities by people, on their well-being. This covariate is divided into three categories (once every month, once every three months, and once every six months = reference). Table 4 contains the supportive analysis to this hypothesis. A person who visits other cities for recreational purposes more frequently, say, once every month ($p$ = 0.008, $\beta$ = –1.16) as opposed to once every six months, is seen to have a higher WHO-5 score, due to the positive $\beta$ coefficient. This is in line with our belief that more frequent visits to other cities for recreational purposes improves their well-being. Further, the likelihood of acquiring mental distress is seen to be lesser for a person who visits other cities for recreational purposes once every month (OR = 0.47, $p$ = 0.0417).

## City specific analysis

Some preliminary city-wise analysis was carried out to find out the distribution of WHO-5 scores across cities and the cohort of people having access to nature (green or blue spaces) around their residences, in each of these cities. Since our sample contained a varied distribution of people from each of the cities, we have shown percentages instead of actual counts for a better appreciation. Fig 4 shows a side-by-side boxplot containing preliminary analysis on how the WHO-5 score is spread across cities. Cities like Ludhiana, Nashik and Vishakhapatnam show a higher WHO-5 score, whereas cities like Agra, Bhopal and Kanpur show a lower score, as compared to others. Fig 5 contains a side-by-side boxplot showing the spread of NIS scores across cities. Ludhiana and Vishakhapatnam show a higher NIS score, as compared to others. Fig 6 contains a stacked barplot showing the percentage of people having access to nature around their residence in each of the 25 cities. As an observation, the top 3 cities where maximum people have access to green and blue spaces happen to be Kochi, Srinagar and Aurangabad. Fig 7 shows the average predicted WHO-5 score for each city as compared to its average nature connectedness score. Eg., Vadodara and Srinagar happen to reflect the lowest NIS scores, resulting in a lower average predicted WHO-5 score, whereas Ludhiana

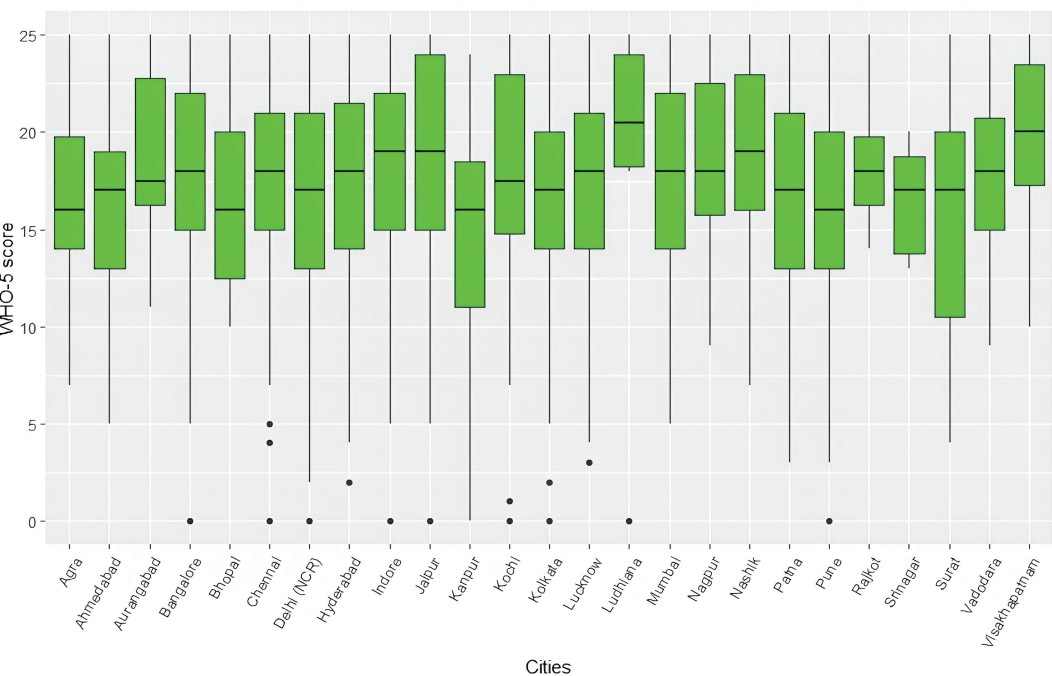

**Fig 4. Side-by-side boxplot showing the distribution of WHO-5 scores in each of the 25 cities.**

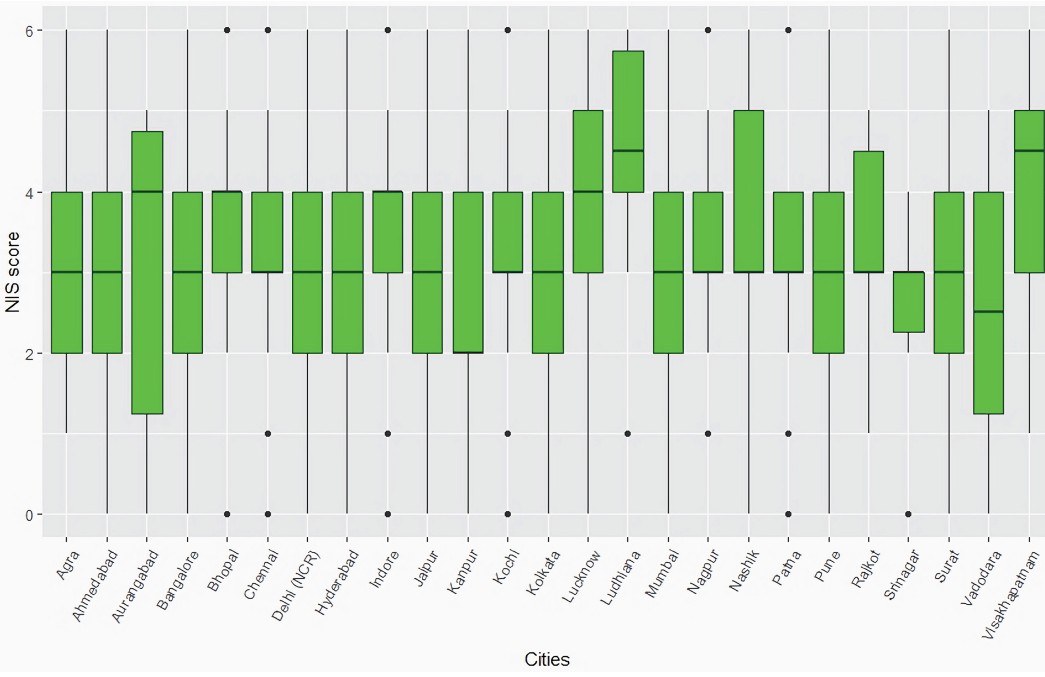

**Fig 5. Side-by-side boxplot showing the distribution of NIS scores in each of the 25 cities.**

shows the highest NIS score, resulting in a higher average predicted WHO-5 score. The four major metro cities namely, Delhi, Mumbai, Chennai and Kolkata reflect in the middle.

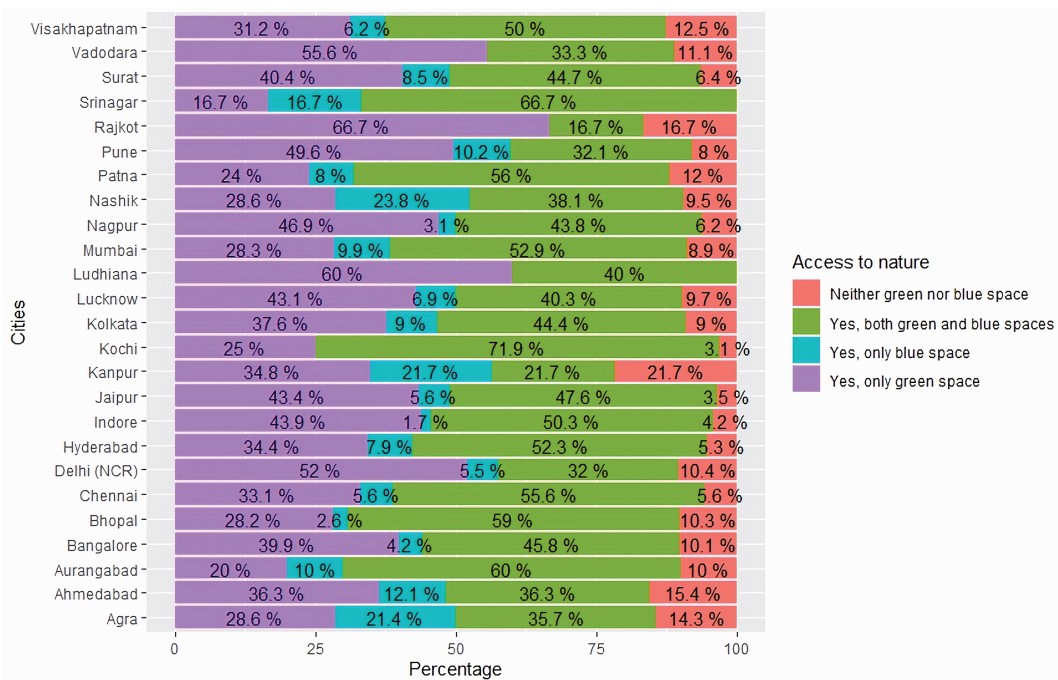

**Fig 6. Stacked barplot showing the percentage of people having access to nature (green or blue spaces) around the residences in each of the 25 cities.**

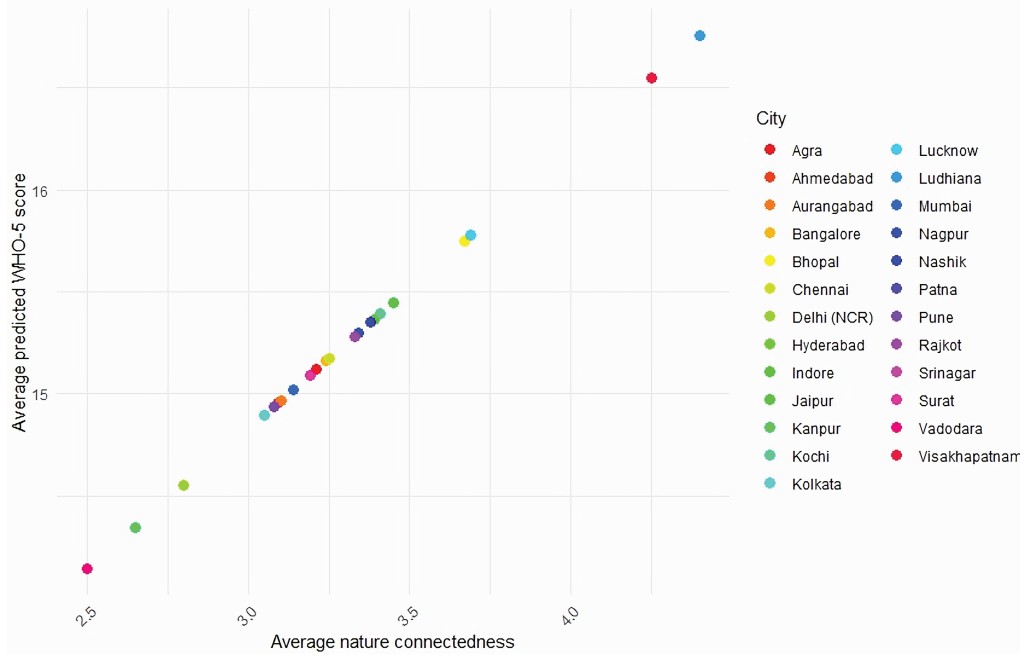

**Fig 7. City level relationship between nature connectedness and well-being.** The plot depicts the predicted WHO-5 score using the MLR model by assuming that all the covariates (age, gender, education, profession, residential exposure, access to green and blue spaces, recreational visits, time spent indoors) are held fixed at their base levels.

## Discussion

This present work is exhaustive as it covers the whole of India, through analysis of the relationship between residential exposure to green and blue spaces and well-being, in the 25 most populated cities of the country.

Supporting hypothesis 1, there was substantial evidence that a residential proximity to a green space had a positive impact on well-being and negative impact on mental distress [57,58]. A green space within 3 Km of the residence had a higher positive impact than that beyond 3 Km. Particularly, a green space situated very close to the residence (within a Km) had a much higher positive impact than a green space located at a distance. A closer proximity to a green space might indeed provide higher motivation to visit such spaces frequently. The same was found in earlier studies [37–39].

However, contrary to the claim, proximity to a blue space was seen to have an opposite impact on well-being and mental distress, than what was expected. Firstly, the number of individuals with residential proximity to blue spaces in urban areas is significantly lower compared to green spaces. The scarcity of blue spaces in many cities, particularly those that are not coastal or lack substantial water bodies such as lakes, ponds, or streams, results in limited exposure for the study participants. As a result, the data corresponding to blue space exposure may be skewed, with a relatively small sample size of individuals benefiting from access to blue spaces.

Moreover, in urban settings, the quality of blue spaces may vary greatly, and proximity alone may not necessarily lead to positive outcomes. For instance, if the blue space is poorly maintained, polluted, or inaccessible for recreational use, it may not have the intended restorative effects on well-being. Additionally, the perception of blue space as a calming and restorative environment may depend on its type and the surrounding urban infrastructure. This contrasts with green spaces, which, on average, tend to be more readily available, accessible, and perceived as beneficial for mental health and well-being. Future studies could explore the quality and types of blue spaces, including water bodies' accessibility, cleanliness, and surrounding areas, to better understand how blue spaces influence well-being and distress in urban youth.

Although blue spaces may generally be considered beneficial for well-being, their limited availability and varied quality in urban environments may explain the unexpected findings in this study. Further research could focus on cities with greater access to blue spaces to offer a more comprehensive understanding of their impact on well-being.

Supporting hypothesis 2, there was significant evidence that opting for nature based recreational visits had a positive impact on well-being and was negatively related to mental distress. Particularly, for people who visited green or blue spaces for recreational purpose less often (once every three months) had a poorer well-being than frequent visitors (once every month). A similar conclusion could be drawn for people visiting other cities for nature based recreation. This suggests that creating more such green spaces (parks, gardens etc.) around residential complexes might be beneficial when it comes to well-being. There should be a norm to construct such spaces (if not available naturally) within the residential complex, benefiting the residents. Additionally, in the Indian context, higher frequency of nature visits has been shown to enhance mindfulness in adults [56], further reinforcing the benefits of such spaces.

Supporting hypothesis 3, there was sufficient evidence that a person who spends more time indoors (between 4 to 12 hours) had a negative impact on well-being as compared to a person who spends less (less than 4 hours). This suggests that in this era of digital advancements, one should not forget to venture out of the house for a casual stroll or undertaking some

household chores such as grocery shopping etc. However, there was no significant evidence of mental distress for a person spending more time indoors.

Supporting Hypothesis 4, substantial evidence indicated that nature connectedness, as measured by the NIS scale, significantly enhanced well-being and reduced mental distress. These results are similar to several studies conducted in Canada, Europe, the United States, Australia, India, Colombia and Hong Kong, which highlighted a strong positive correlation between nature connectedness and well-being [39].

This study advances the theoretical understanding of the relationship between nature connectedness and well-being in urban environments, offering new insights into how proximity and engagement with nature influence psychological outcomes. By differentiating the effects of green and blue spaces, this study refines existing conceptual models, demonstrating that accessibility alone is not sufficient; quality, usability, and individual perception play a crucial role. The findings underscore that while proximity to green spaces consistently improves well-being, blue spaces may not yield similar benefits due to variations in availability, maintenance, and recreational access. This contributes to a more nuanced framework for understanding the impact of nature connectedness on well-being and mental distress among urban youth, highlighting the importance of both physical access and subjective experience of being in nature on well-being.

Additionally, this study provides empirical validation of nature connectedness as a significant psychological construct within an Indian urban context. While much of the existing literature on nature connectedness and well-being originates from Western settings, this research extends these insights to a densely populated, rapidly urbanizing society. By linking nature connectedness with mental health benefits beyond mere exposure such as through active recreational engagement, this study emphasizes the role of behavioural interaction with nature in enhancing well-being. These findings lay the groundwork for future interdisciplinary research, integrating psychology, urban planning, and public health to develop holistic strategies that enhance mental resilience in urban populations.

## Comparative analysis with domestic and international studies

The findings of this study align with a growing body of international research highlighting the positive relationship between nature connectedness and mental well-being. A meta-analysis covering multiple studies from Canada, Europe, the United States, Australia, India, Colombia, and Hong Kong has shown that individuals with higher nature connectedness experience better psychological well-being, particularly in terms of personal growth and eudaimonic well-being [39]. Consistent with these findings, our study reaffirms that nature connectedness is a significant predictor of well-being among urban youth in India, further supporting the cross-cultural validity of this relationship. Notably, research from Canada [47] and the United States [48] suggests that nature connectedness is a distinct and robust predictor of happiness, while studies in Australia [51] and the UK [50,52] emphasize that engagement with nature has a stronger impact on well-being than merely spending time outdoors. These insights align with our study's emphasis on active and meaningful interaction with nature rather than passive exposure.

However, a key divergence from international research is the lack of a significant positive association between proximity to blue spaces and well-being in the Indian urban context. While studies across Europe, North America, and Australia have consistently reported the restorative benefits of blue spaces (attributing them to recreational opportunities, scenic value, and water quality) [49] our findings suggest that such benefits may not be universally applicable. This discrepancy could be due to limited access to well-maintained blue spaces in

these large urban Indian cities, differences in cultural attitudes toward water bodies among urban youth, or environmental concerns such as pollution and overcrowding. Moreover, international studies have highlighted how simple nature-related activities, such as caring for houseplants in China [53] or spending time in private green spaces in England [55], enhance mindfulness and mental well-being. Similarly, Indian research suggests that frequent nature visits enhance mindfulness, boost subjective happiness, and build resilience. By situating our study within this broader international discourse, we contribute to a more nuanced understanding of how nature connectedness might function in different socio-cultural contexts, emphasizing the need for localized strategies to enhance urban well-being.

## Practical implications

Our study, focusing on urban youth, underscores the positive relationship between connection with nature and well-being, which has significant implications for urban planning, public mental health policies, and community interventions. Urban youth, particularly in fast-paced, high-stress environments, are increasingly experiencing mental health challenges due to factors like pollution, limited access to nature, and social isolation. In this context, enhancing the availability and accessibility of green and blue spaces becomes crucial. Urban planners should prioritize creating parks, gardens, and water bodies within cityscapes, ensuring that these spaces are not only available but also equitably distributed across neighbourhoods, providing all urban youth with the opportunity to connect with nature. Designing restorative urban environments—such as tree-lined streets, green rooftops, and nature corridors—can further help mitigate urban stressors and promote well-being for young people navigating the complexities of city life.

In terms of public mental health policies, our study suggests that integrating nature-based interventions can play a vital role in promoting mental health and well-being among urban youth. Public health campaigns could include nature walks, forest therapy programs, or outdoor community activities tailored to engage young people and reduce stress. Policies should advocate for the preservation and expansion of green spaces within urban areas, viewing nature not just as a luxury, but as a crucial element for the psychological well-being of youth. On the community level, interventions such as urban gardening, eco-volunteering, and nature-based recreational programs can foster a stronger sense of community and connection to the environment. These initiatives can serve as both mental health interventions and tools for youth empowerment, giving them the opportunity to engage meaningfully with their surroundings. Additionally, educational programs in schools and community centres can raise awareness about the mental health benefits of nature, encouraging young people to regularly engage with green spaces. These practical applications of our research offer a clear path forward to improve the mental health and well-being of urban youth, creating healthier, more resilient communities in urban environments.

## Limitations and recommendations for future research

While this study represents a significant and comprehensive exploration of the relationship between nature connectedness and well-being among urban youth in India, certain limitations must be acknowledged. These limitations offer avenues for improvement in future research.

Firstly, the study relied on self-reported survey data, which tends to carry potential biases such as social desirability bias or inaccuracies in recall. While measures like ensuring anonymity and using validated scales helped mitigate these risks, future studies could

incorporate objective measures to strengthen the validity of findings. For instance, leveraging participants' Google Maps location history or wearable device metrics could provide more precise data on exposure to green and blue spaces and time spent outdoors versus indoors.

Secondly, the study sample was drawn from the 25 most populated cities in India, with an uneven distribution of responses across cities. For example, response counts ranged from as few as six in Rajkot to 415 in Delhi NCR, reflecting significant disparity. This uneven distribution could impact the generalizability of the results. Future research could address this by employing systematic or stratified sampling techniques to ensure more balanced representation across locations.

Thirdly, the cross-sectional design of the study limited the ability to establish causal relationships between nature connectedness and well-being. Longitudinal studies, which track participants over time, could provide stronger evidence of causation and capture temporal changes in behaviour and experiences. Additionally, qualitative methods like interviews or focus groups, which were not feasible in this study due to budgetary constraints, could complement quantitative findings and offer richer insights into how urban youth perceive and engage with nature.

Seasonality was another factor that the current study did not account for. Since India experiences distinct seasonal variations, exposure to nature and its effects on well-being may vary significantly across seasons. For instance, individuals may spend more time outdoors in winter compared to summer, which could influence their connection with nature. Future research should aim to collect data across different seasons to better understand these variations and their potential impact.

While our study includes a range of covariates, it may not account for all potential confounding factors that could influence well-being. In our analysis, we included key covariates such as age, gender, education level, and geographical location, which are known to affect both nature connectedness and well-being based on existing literature. However, we recognize that other factors, such as socio-economic status, pre-existing mental health conditions, and environmental influences unrelated to nature, could also play a significant role in shaping well-being. Due to data availability and scope limitations, these additional factors were not included in the current study. We suggest that future research could expand the range of covariates to better capture the full complexity of influences on well-being.

Future research may also explore potential mediating and moderating variables that influence the relationship between nature connectedness and well-being. Mediating factors such as mindfulness, social interaction, and physical activity could help explain how nature connectedness leads to improved well-being. For instance, spending time in nature may enhance mindfulness, which in turn reduces stress and enhances resilience. Similarly, moderating variables like personality traits, accessibility, and cultural attitudes toward nature could determine for whom or under what conditions nature connectedness might have the strongest effect. Examining these variables through structural equation modelling or moderated mediation analyses could offer deeper insights into the mechanisms underlying this relationship and inform more targeted interventions for diverse urban populations.

Furthermore, the study focused on a narrow age group of 18–25 years, limiting its applicability to young adults exclusively. Extending the sample to include individuals up to 30 years of age, encompassing young professionals, could provide a broader understanding of the relationship between nature connectedness and well-being.

Although the study emphasized statistically significant results, the practical significance and effect sizes of the findings warrant further exploration. Future analyses could place greater focus on effect sizes to better assess the strength and real-world relevance of the observed relationships.

Future research should aim to replicate this study at both local and global levels to enhance the robustness and generalizability of the findings. Within India, city-specific studies can provide deeper insights into how regional differences in urban infrastructure, cultural attitudes toward nature, and socio-economic factors influence the relationship between nature connectedness and well-being. Longitudinal studies tracking changes over time could help establish causal relationships, while expanding the sample to include diverse age groups and socio-economic backgrounds would further validate the findings. Collaborations with local universities and research institutions can facilitate these efforts, ensuring that findings are contextually relevant and applicable to regional urban planning and mental health policies.

On a global scale, similar research can be conducted in rapidly urbanizing regions, particularly in Global South nations facing similar challenges in urban expansion, green space accessibility, and public health. Comparative studies with developed nations can help identify cultural and infrastructural differences in how urban nature affects well-being. Standardized psychological measures and survey instruments can be utilized to ensure cross-cultural comparability. Additionally, future research should explore qualitative dimensions of urban nature, such as perceived accessibility, safety, and usability, rather than relying solely on proximity measures. This will help refine existing models of nature connectedness and its psychological benefits.

Despite these limitations, this study offers valuable insights into the role of nature connectedness in promoting mental well-being among urban youth in India. It highlights key implications for urban planning and mental health promotion. Policymakers can use these findings to prioritize the inclusion of accessible green and blue spaces, such as parks, urban forests, and water bodies in urban development projects. These spaces should be strategically located near residential areas to encourage regular use and engagement.

Moreover, merely increasing access to nature may not suffice. Active engagement through nature-based interventions could amplify the benefits. Initiatives such as forest bathing workshops, mindfulness or yoga sessions in natural spaces, and community gardening projects can help individuals develop a stronger connection with nature. Schools, colleges, and workplaces could collaborate to implement programs like outdoor learning activities or gamified challenges to explore natural spaces.

By combining the creation of accessible natural environments with targeted interventions, urban youth can be encouraged to actively interact with nature, fostering improved well-being and reduced mental distress. Future research should continue building on these findings to deepen our understanding and enhance practical applications for urban planning and public health.

## Acknowledgements

The authors would like to thank Juhi Goyal for her research assistance.

## Author contributions

**Conceptualization:** Raina Chhajer, Sudeep Bapat.

**Data curation:** Raina Chhajer, Sudeep Bapat.

**Formal analysis:** Sudeep Bapat.

**Investigation:** Raina Chhajer, Sudeep Bapat.

**Methodology:** Raina Chhajer, Sudeep Bapat.

**Project administration:** Raina Chhajer.

**Visualization:** Raina Chhajer, Sudeep Bapat.

**Writing – original draft:** Raina Chhajer, Sudeep Bapat.

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
