## [Decision Letter · Decision Letter 0]

1 Nov 2024

PONE-D-24-25074Exploring the impact of nature connectedness on well-being among urban youth from 25 most populated cities in IndiaPLOS ONE

Dear Dr. Bapat,

Thank you for submitting your manuscript to PLOS ONE. After careful consideration, we feel that it has merit but does not fully meet PLOS ONE’s publication criteria as it currently stands. Therefore, we invite you to submit a revised version of the manuscript that addresses the points raised during the review process.

We look forward to receiving your revised manuscript.

Kind regards,

Weicong Li, P.hD

Academic Editor

PLOS ONE

Journal Requirements:

4. In this instance it seems there may be acceptable restrictions in place that prevent the public sharing of your minimal data. However, in line with our goal of ensuring long-term data availability to all interested researchers, PLOS’ Data Policy states that authors cannot be the sole named individuals responsible for ensuring data access (http://journals.plos.org/plosone/s/data-availability#loc-acceptable-data-sharing-methods).

6. We note that Figure 1 in your submission contain map images which may be copyrighted. All PLOS content is published under the Creative Commons Attribution License (CC BY 4.0), which means that the manuscript, images, and Supporting Information files will be freely available online, and any third party is permitted to access, download, copy, distribute, and use these materials in any way, even commercially, with proper attribution. For these reasons, we cannot publish previously copyrighted maps or satellite images created using proprietary data, such as Google software (Google Maps, Street View, and Earth). For more information, see our copyright guidelines: http://journals.plos.org/plosone/s/licenses-and-copyright.

1) You may seek permission from the original copyright holder of Figure 1 to publish the content specifically under the CC BY 4.0 license.  

2) If you are unable to obtain permission from the original copyright holder to publish these figures under the CC BY 4.0 license or if the copyright holder’s requirements are incompatible with the CC BY 4.0 license, please either i) remove the figure or ii) supply a replacement figure that complies with the CC BY 4.0 license. Please check copyright information on all replacement figures and update the figure caption with source information. If applicable, please specify in the figure caption text when a figure is similar but not identical to the original image and is therefore for illustrative purposes only.

7. Please upload a new copy of Figures 3 and 4 as the detail is not clear. Please follow the link for more information: 

https://blogs.plos.org/plos/2019/06/looking-good-tips-for-creating-your-plos-figures-graphics/

https://blogs.plos.org/plos/2019/06/looking-good-tips-for-creating-your-plos-figures-graphics/

**Additional Editor Comments: **

Please note that I have acted as a reviewer for this manuscript, and you will find my comments below, under Reviewer 4.

The authors need to address the following issues:

1. please elaborate on the sample selection strategy in the Methods section and explore the impact of this distribution on the findings in the Discussion section.

2. strengthen the theoretical support for the hypotheses, particularly by discussing the basis for the hypotheses in the context of the existing literature.

3. explain the data collection process and bias control measures in more detail in the Methods section.

4. insufficient explanation of the negative correlation between blue space and mental health.

5. discuss the limitations of cross-sectional study design and uneven sample distribution.

6. Add a comparative analysis with international studies to clarify the unique contribution of this study in a global context.

7. Define the scope of “natural connections” and explain why these cities were selected and their value for other urban studies.

8. Add specific recommendations for urban planning and policy applications and expand the literature review to include more indigenous Indian studies and regional data.

Reviewers' comments:

Reviewer's Responses to Questions

**Comments to the Author**

1. Is the manuscript technically sound, and do the data support the conclusions?

Reviewer #1: Yes

Reviewer #2: Yes

Reviewer #3: Partly

Reviewer #4: Partly

2. Has the statistical analysis been performed appropriately and rigorously? 

Reviewer #1: Yes

Reviewer #2: Yes

Reviewer #3: N/A

Reviewer #4: N/A

3. Have the authors made all data underlying the findings in their manuscript fully available?

Reviewer #1: Yes

Reviewer #2: Yes

Reviewer #3: No

Reviewer #4: Yes

4. Is the manuscript presented in an intelligible fashion and written in standard English?

Reviewer #1: Yes

Reviewer #2: Yes

Reviewer #3: Yes

Reviewer #4: Yes

5. Review Comments to the Author

Reviewer #1: I reviewed the article titled "Exploring the impact of nature connectedness on well-being among urban youth from 25 most populated cities in India". It contributes significantly to the scientific literature in several ways:

1- This study addresses the lack of research on the relationship between nature connectedness and well-being among urban youth in India, a region where such studies are scarce. By focusing on a large and diverse sample from 25 major cities, the research provides a comprehensive understanding of how nature connectedness influences the mental health of urban youth in a rapidly urbanizing country.

2- The study employs robust statistical methods, including multivariate and logistic regression analyses, to explore the impact of residential exposure to green and blue spaces, frequency of recreational visits, and time spent indoors on well-being. This multifaceted approach offers nuanced insights into the various factors that contribute to mental health outcomes.

3- The findings have significant implications for urban planning and public health policies. By demonstrating the positive effects of proximity to green and blue spaces and frequent interaction with nature on mental well-being, the study advocates for the development of more parks and green spaces in urban areas. This evidence can be used to inform policies aimed at improving the mental health of urban populations, particularly the youth.

4- While the study is region-specific, its findings contribute to the broader global literature on the benefits of nature connectedness. By providing empirical evidence from a developing country context, the research adds a valuable perspective to the existing knowledge, which is predominantly based on studies from Western countries.

5- The study’s results can be used by educators, mental health professionals, and urban planners to design interventions that promote nature connectedness as a means of enhancing mental well-being among youth. This practical application further underscores the importance of the research.

Based on the review of the study, below are the overall weaknesses identified which need the author(s) attention:

1- The study is based on responses from 25 cities, but the number of responses per city is uneven, ranging from 6 in Rajkot to 415 in Delhi. This uneven distribution could lead to sampling bias and may not accurately represent the youth populations in all cities.

2- The study uses a cross-sectional design, which limits the ability to draw causal inferences. It only captures a snapshot in time rather than observing changes over time, which could provide more robust evidence for the relationships explored.

3- The data relies entirely on self-reported measures, which can be prone to biases such as social desirability bias and recall bias, potentially leading to inaccurate reporting of nature connectedness and well-being.

4- Since the data collection was conducted through an online survey, there is no way to validate the accuracy of the responses, leading to potential inaccuracies in the data.

5- Some survey questions are described as ambiguous, which could lead to misinterpretation by participants and affect the validity of the results.

6- The study focuses only on youth aged 18-25 years. This narrow age range limits the generalizability of the findings to other age groups, such as younger adolescents or older adults.

7- The study is limited to urban areas in India, and the findings may not be generalizable to rural areas or other countries with different cultural and environmental contexts.

8- The study does not account for seasonal variations, which could affect outdoor activities and nature connectedness. For instance, participants might engage more with nature during certain seasons, influencing their well-being.

9- The study finds that proximity to blue spaces has an unexpected negative impact on well-being, but this result is not deeply explored. The lack of blue spaces in the sample could have skewed the data, but this issue is not adequately addressed.

10- While the study includes various covariates, it may not account for all potential confounding factors, such as socio-economic status, existing mental health conditions, or environmental factors unrelated to nature that could influence well-being.

11- Although some limitations are acknowledged, the discussion might not be comprehensive enough, leaving out other potential weaknesses that could impact the study’s validity.

12- The study relies heavily on quantitative data, with little to no qualitative insights that could provide a deeper understanding of the subjective experiences of nature connectedness and its impact on well-being.

13- With 2,283 participants responding to an online survey, there is a risk of survey fatigue, where participants may not fully engage with the later questions, leading to incomplete or less thoughtful responses.

14- The study may focus heavily on statistically significant results without thoroughly considering the practical significance or the effect sizes, which might be small even when statistically significant.

In addition to the weaknesses mentioned above, below are suggested area of improvement divided section by section:

1- Title : Consider rephrasing the title to include the terms "urban mental health" or "youth well-being" to better capture the essence of the study.

2- Abstract: Add a brief mention of key limitations in the abstract to provide a more balanced overview of the study’s contributions and its constraints.

3- Introduction: Expand the discussion on how this study differs from or builds upon previous research in other countries. This will help clarify the novelty of the study and its contribution to the global literature. Provide a more detailed explanation of how each hypothesis is derived from the theoretical framework or past studies to strengthen the foundation of the research.

4- Materials and Methods: Acknowledge the sampling bias more explicitly in this section and discuss potential strategies for mitigating this issue in future studies, such as stratified sampling or oversampling underrepresented cities. Include a discussion on the cultural relevance of the WHO-5 and NIS scales in the Indian setting and any modifications made to ensure their validity and reliability. Suggest alternative or complementary data collection methods, such as observational data or longitudinal tracking, to strengthen the study's validity and reduce reliance on self-report. Provide a more detailed justification for the inclusion of specific covariates and discuss any additional variables that might have been considered.

5- Results: Include more context or commentary on the significance of the descriptive statistics, especially how they relate to the urban youth population in India. Expand the discussion on the counterintuitive results, such as the negative association with blue spaces, by exploring potential cultural or environmental factors specific to the Indian context. Discuss the limitations of the city-specific analysis due to uneven sample sizes and consider recommending further research to validate these findings with more balanced samples.

6- Discussion: Provide a more in-depth discussion of the limitations, particularly the sampling bias and the reliance on cross-sectional data, and suggest ways these could be addressed in future research. Include a more comprehensive comparison with international studies, highlighting both the similarities and differences, and discuss the implications for cross-cultural research on nature connectedness. Provide more concrete examples of how urban planners and policymakers can use the findings to design interventions that promote nature connectedness among urban youth.

7- Conclusion: Streamline the conclusion to emphasize the key contributions and their implications for both the scientific community and policymakers. Propose specific future research directions that address the study’s limitations, such as exploring seasonal variations, using a longitudinal design, or incorporating qualitative methods.

Reviewer #2: The manuscript demonstrates technically sound scientific research, with data to support the conclusions presented. The experiments were conducted rigorously, using appropriate controls, replication and adequate sample sizes, allowing conclusions to be drawn appropriately based on the data presented.

The conclusions are carefully drawn from the data presented and corroborate the initial hypotheses. The study shows that residential proximity to green spaces, frequent recreational visits to these spaces and a strong connection with nature are significantly associated with better mental well-being, while spending more time indoors has a negative impact on mental health. some pointers to improve the manuscript can be found in the attached documents

Reviewer #3: This manuscript explores an interesting question—the relationship between nature connectedness and the mental health of urban adolescents in India. From my personal perspective, considering the accelerated process of urbanization and the exacerbation of adolescent mental health issues in the post-pandemic period, this study holds certain practical significance.

1. Although the study is methodologically well-structured and employs multiple regression and logistic regression models, there are some shortcomings in the hypothesis section. The research could strengthen its hypotheses by providing more detailed theoretical foundations. Furthermore, the discussion on the negative correlation between blue spaces (e.g., bodies of water) and mental health is rather superficial. It is recommended to delve deeper into the possible reasons for this result in the discussion section, such as the impact of urban geographical location, urban planning, or other socio-cultural factors, and to cite more relevant literature to support this discussion. This would make the conclusions more reasonable and convincing.

2. The study is based on self-reported survey data, which introduces potential biases and challenges in validation. As the authors have mentioned, the sample distribution across different cities is uneven, and the sample size in some cities is too small, which may affect the generalizability of the research results. Uneven sample distribution among different cities may impact the generalizability of the results, especially in cities with extremely small sample sizes. It is suggested to explain the sample selection strategy in detail in the methods section and to further discuss in the limitations how this issue affects the research results. Future research could consider allocating samples more evenly across different cities.

3. Since this manuscript is based on self-reported data, there may be biases. Incorporating more indigenous studies on the mental health of urban adolescents in India and their connection with nature would further enhance the background and relevance of the research. Although the citation of global studies is sufficient, the article could improve its quality by integrating more regional data and research. For example, providing more explanations about the data collection process, questionnaire design, and how biases were controlled, and including key questions from the questionnaire in the appendix, would further enhance the transparency of the data and the scientific rigor of the research.

Reviewer #4: 1. Please define the scope of 'natural connection', what does it include? This will help the understanding of scholars not in the field.

2. please provide justification for the choice of this city.

3. is the study of India useful for urban studies in other countries?

6. PLOS authors have the option to publish the peer review history of their article (what does this mean?). If published, this will include your full peer review and any attached files.

Reviewer #1: **Yes: **Abdullah Addas

Reviewer #2: No

Reviewer #3: No

Reviewer #4: No

---

## [Author Response · Author response to Decision Letter 1]

12 Dec 2024

Attached as a separate file named "Response to reviewers".

---

## [Decision Letter · Decision Letter 1]

11 Feb 2025

PONE-D-24-25074R1Exploring the impact of nature connectedness on urban mental health and well-being among youth from 25 most populated cities in IndiaPLOS ONE

Dear Dr. Bapat,

Thank you for submitting your manuscript to PLOS ONE. After careful consideration, we feel that it has merit but does not fully meet PLOS ONE’s publication criteria as it currently stands. Therefore, we invite you to submit a revised version of the manuscript that addresses the points raised during the review process.

We look forward to receiving your revised manuscript.

Kind regards,

Weicong Li, P.hD

Academic Editor

PLOS ONE

 Journal Requirements:

Additional Editor Comments :

Please find my comment in reviewer 4. The manuscript has research value, but still needs to address the following major issues. See my comments.

Reviewers' comments:

Reviewer's Responses to Questions

**Comments to the Author**

1. If the authors have adequately addressed your comments raised in a previous round of review and you feel that this manuscript is now acceptable for publication, you may indicate that here to bypass the “Comments to the Author” section, enter your conflict of interest statement in the “Confidential to Editor” section, and submit your "Accept" recommendation.

Reviewer #2: (No Response)

Reviewer #4: (No Response)

2. Is the manuscript technically sound, and do the data support the conclusions?

Reviewer #2: Yes

Reviewer #4: Partly

3. Has the statistical analysis been performed appropriately and rigorously? 

Reviewer #2: Yes

Reviewer #4: N/A

4. Have the authors made all data underlying the findings in their manuscript fully available?

Reviewer #2: Yes

Reviewer #4: Yes

5. Is the manuscript presented in an intelligible fashion and written in standard English?

Reviewer #2: No

Reviewer #4: Yes

6. Review Comments to the Author

Reviewer #2: the authors responded to requests to adjust the text with significant additions and improvements to the work.

Reviewer #4: 1. Please improve the quality of all Figures. the text in Figure 8 is too small, making it difficult to read the content.

2. “age”, “gender”, “education”, and “profession” in the bar chart can be used as subfigures to form a completed chart.

3. Please check the spelling of the second author in References 59.

4. It is recommended that an academic contribution map be drawn in “Discussions” to illustrate the contribution of the research to the field as a whole.

5. Also, in Discussions, explain how the research will be replicated and disseminated. Include the study in the local area, as well as the impact on the global scale.

6. Integrate theoretical frameworks, such as the Biophilia Hypothesis or Attention Restoration Theory, in the Introduction section as theoretical support for subsequent research.

7. Refine the logical relationship between “natural connectivity” and “mental health” and explain its possible mediating or moderating variables.

8. How is the variable “indoor time” defined and measured? The authors may add to the list.

Compare and contrast the results of domestic and international studies, analyze the possible reasons behind the differences, and highlight the academic contributions of this study.

7. PLOS authors have the option to publish the peer review history of their article (what does this mean?). If published, this will include your full peer review and any attached files.

Reviewer #2: No

Reviewer #4: **Yes: **Weicong Li

---

## [Author Response · Author response to Decision Letter 2]

27 Mar 2025

Kindly note that we have changed the title a bit as we think it fits better to the current study, focusing on "urban youth". All other responses to reviewers' comments are included as a separate file.

---

## [Editor Report · Decision Letter 2]

14 Apr 2025

Exploring the impact of nature connectedness on well-being and mental distress among urban youth: Evidence from 25 most populated cities in India

PONE-D-24-25074R2

Dear Dr. Bapat,

We’re pleased to inform you that your manuscript has been judged scientifically suitable for publication and will be formally accepted for publication once it meets all outstanding technical requirements.

Kind regards,

Weicong Li, P.hD

Academic Editor

PLOS ONE
---

## [Editor Report · Acceptance letter]

PONE-D-24-25074R2

PLOS ONE

Dear Dr. Bapat,

I'm pleased to inform you that your manuscript has been deemed suitable for publication in PLOS ONE. Congratulations! Your manuscript is now being handed over to our production team.

Kind regards,

on behalf of

Dr. Weicong Li

Academic Editor

PLOS ONE